# Absence of *Figla-like* Gene Is Concordant with Femaleness in Cichlids Harboring the LG1 Sex-Determination System

**DOI:** 10.3390/ijms23147636

**Published:** 2022-07-11

**Authors:** Arie Yehuda Curzon, Andrey Shirak, Ayana Benet-Perlberg, Alon Naor, Shay Israel Low-Tanne, Haled Sharkawi, Micha Ron, Eyal Seroussi

**Affiliations:** 1Institute of Animal Science, Agricultural Research Organization, Rishon LeTsiyon 75288, Israel; arie.curzon@mail.huji.ac.il (A.Y.C.); shiraka@volcani.agri.gov.il (A.S.); micha@agri.huji.ac.il (M.R.); 2Robert H. Smith Faculty of Agriculture, Food and Environment, Hebrew University of Jerusalem, Rehovot 76100, Israel; 3Dor Research Station, Division of Fishery and Aquaculture, Hof HaCarmel 30820, Israel; ayanab@moag.gov.il (A.B.-P.); alonn@moag.gov.il (A.N.); shaiis@moag.gov.il (S.I.L.-T.); haleds@moag.gov.il (H.S.)

**Keywords:** sex determination, *figla-like*, cichlids, tilapia, master key regulator

## Abstract

*Oreochromis niloticus* has been used as a reference genome for studies of tilapia sex determination (SD) revealing segregating genetic loci on linkage groups (LGs) 1, 3, and 23. The master key regulator genes (MKR) underlying the SD regions on LGs 3 and 23 have been already found. To identify the MKR in fish that segregate for the LG1 XX/XY SD-system, we applied short variant discovery within the sequence reads of the genomic libraries of the *Amherst* hybrid stock, *Coptodon zillii* and *Sarotherodon galilaeus*, which were aligned to a 3-Mbp-region of the *O. aureus* genome. We obtained 66,372 variants of which six were concordant with the XX/XY model of SD and were conserved across these species, disclosing the male specific *figla-like* gene. We further validated this observation in *O. mossambicus* and in the *Chitralada* hybrid stock. Genome alignment of the 1252-bp transcript showed that the *figla-like* gene’s size was 2664 bp, and that its three exons were capable of encoding 99 amino acids including a 45-amino-acid basic helix–loop–helix domain that is typical of the ovary development regulator—*factor-in-the-germline-alpha* (*FIGLA*). In *Amherst* gonads, the *figla-like* gene was exclusively expressed in testes. Thus, the *figla-like* genomic presence determines male fate by interrupting the female developmental program. This indicates that the *figla-like* gene is the long-sought SD MKR on LG1.

## 1. Introduction

Genetic sex determination (SD) is principally driven by a single gene, a master key regulator (MKR), capable of turning on an alternative developmental program to that maintained in the homogametic state (XX or ZZ) [1,2,3]. Among vertebrates, fishes exhibit the most diverse SD mechanisms including genetic and environmental SD [4,5,6]. In most mammals, the same MKR gene controls an XX/XY SD system, i.e., the sex-determining region Y (SRY) [7,8]. Similarly in birds, a Z-linked *dmrt1* is the common MKR utilized for the WZ/ZZ SD system [9,10]. However, different fish species have adopted different MKR genes to initiate the sex cascade, which determines the fate of the bipotential sex gonad [11]. During the last year, two new MKRs (*bmpr1b* and *banf2w*) have been suggested [12,13], in addition to ten SD MKRs detected previously [11]. In most cases in fishes, as in other vertebrates, Y and W differ from X and Z, respectively, in the number of copies and sequences [11,14]. Even for closely related fish species such as of the *Oreochromis* and *Oryzias* genera, different MKRs have been identified, thus elucidating the complexity of the SD network [15,16].

Showing high diversification of SD systems including XX/XY and WZ/ZZ genetic systems and environmental SD, African cichlids have gained interest as a model for adaptive radiation evolution [16,17]. Tilapia is considered the most economically important clade among cichlids. However, commercial farming of tilapia requires all-male progeny (AMP). This can be achieved by feeding fry with synthetic analogs of androgens [18,19], although, hormone use could constitute a hazard for health and the environment [18,20]. An alternate method for AMP is the breeding of YY “super males”, which can be performed by mating XY females from a sex-reversed population, with XY males [21,22]. However, this technology was not effective in our experiments, which utilized purebred species, due to the significantly lower sexual activity of YY males in comparison with XY males (data not shown). An alternative approach in tilapia is hybridization of different purebred *Oreochromis* species, such as *O. aureus* (*Oa*) males with *O. niloticus* (*On*) females [23,24,25]. Such crosses are defined as “all-male crosses” as they result in AMP, and the mechanism is assumed to be driven by the variability in the SD mechanisms of the parents. However, as in the case of the YY “super males” approach, this alternative was not practical for mass production because of low reproductive interactions between males and females of different species [19,26,27]. The mating of hybrids is an alternative for AMP in tilapia; by using fish of different SD systems, the reproductive behavioral barrier is avoided. Studies have reported crosses between *Oa* males and *On* females that have yielded hybrid couples, which reproduced under natural conditions and produced AMP [28]. The same mating scheme has been suggested through analysis of admixed stocks [19,29]. These results have indicated that the number of loci involved in the SD of *Oa* × *On* hybrids is restricted, and that the allelic patterns of the SD loci can be restored to that of the original purebred species, while maintaining reproductive interaction. Hence, understanding the SD mechanisms in tilapia purebred species and their hybrids is essential for the production of sustainable hormone-free AMP [19,28,29].

In general, purebred species of tilapia possess monofactorial SD systems such as WZ/ZZ and XX/XY [30,31]. Nevertheless, *On*, *Oa*, and their hybrids have been reported to segregate for three SD loci on LGs 1 [32,33], 3 [12,34], and 23 [14,35,36]. Purebred *Oa* and *On* have been reported to segregate for the SD systems on LGs 3 (WZ/ZZ) and 23 (XX/XY), respectively. Yet, the XX/XY SD system on LG1 was suggested to be a result of hybridization [14,19,37], which is in accordance with the autosomal theory suggesting that loci that were in the homozygous autosomal state of progenitors do segregate in hybrid offspring [38]. Another possibility is that the SD on LGs 1 and 3 have been introduced by additional *Oreochromis* species, due to contamination caused through aquaculture commercial processes and breeding programs [39,40,41,42]. However, some oppose the basic hypothesis that the SD is monofactorial in purebred tilapia species [39,40,41,43].

An additional complexity in cichlids is their equivocal taxonomy. As an example, *Oa* and *Sarotherodon galilaeus* (*Sg*), which are classified as two different tilapia genera, have a mitochondrial *cox1* sequence difference lower than the minimal interspecies threshold of 1% [44,45,46,47,48,49]. Additionally, significant discrepancies have been detected in tilapias between mitochondrial and genomic phylogeny [50]. Three groups of cichlids (*Oreochromis*, *Sarotherodon*, and *Coptodon*) are classified based on spawning behavior and the way in which they carry fertilized eggs and embryos [51,52,53]. This classification does not always match barcoding by mitochondrial sequences, which is widely used for species classification.

The study of the XX/XY system on LG1 has become the principal focus of SD studies in tilapia [32,33,37,39,42,54,55,56,57,58,59,60,61]. Yet, the MKR underlying the SD region has not been discovered. This is in contrast with the two other MKR on LGs 3 and 23 that have been identified, i.e., *barrier-to-autointegration-factor 2* (*banf2*) [12] and *anti-Müllerian hormone* (*amh*), respectively [14,35,36]. LG1 has gained interest as a unique SD locus that is involved in the SD of both hybrids of *Oa* × *On* [14,29] and of three purebred species *O. mossambicus (Om)*, *C. zillii (Cz)* [57,62,63], and *S. melanotheron* (*Sm*) [54]. In contradiction to this, *O. mossambicus* was lately found to have an SD locus segregating on LG14 [57]. However, in a previous study, we found that the definition of *Om* is ambiguous, as two Barcode Index Numbers (BIN) identifiers in the BOLD taxonomy dataset are defined as *Om* [64]. One *Om* form (*OmI*), defined in the BOLD system (https://www.boldsystems.org, accessed on 10 July 2022) as BOLD: ADI0792, is currently maintained in the Agricultural Research Organization (Israel) and was established from fish, which originated in Natal (RSA) [65]. In *OmI*, the SD locus segregates on LG1 [62,63]. Yet, another form of *Om* (*OmII*), defined as BOLD: AAA8511, differs by 4% of the *cox1* sequence from *OmI*. Undoubtedly, these sequence differences in *cox1* barcodes indicate that the two forms of *Om* are in fact two different species [64] with two different segregating SD loci on different linkage groups (LGs 1 and 14).

The SD locus on LG1 has been detected by linkage and association studies of independent groups in a proximate similar region on the current genome build of *On* (Genome accession number: GCA_001858045.3), 24–27 Mbp (Figure 1) [19,29,42,58,60,61]. Initially, the SD region spanned between the GM201 and UNH995 microsatellite markers (Figure 1) [42]. Placed in this region, *wt1b* was suggested as a candidate SD MKR. Nonetheless, this candidacy was later rejected, as it was proven that *wt1b* was outside the narrowed SD region [58]. Using RAD sequencing, a few sex-linked SNPs (Oni61067, Oni23063, and Oni28137) were found by additional studies (Figure 1) [60,61]. Our group found that the two microsatellite markers BYL018 [19,29] and BYL012 (developed by Dr. Bo-Young Lee in Prof T.D. Kocher’s Lab) were almost completely linked with sex in fish stocks, which segregate for LG1 only. Whole genome sequencing (WGS) and gonad transcriptome are available for descendants of a commercial *On* strain from Amhrest (*As*), which segregates on LG1 [32,33]. The *On* genomic map has been available for use as a reference for the study of LG1 for many years [17], whereas the *Oa* genome has only recently been published [34].

The failure to detect an MKR may be related to the reference genome utilized. It is well-established that LG1 segregates in *O. aureus* × *O*. *niloticus* hybrids [14,29]. To find the LG1 MKR that segregates in multiple cichlid species and their hybrids, in this study, we chose to analyze the critical SD region on LG1, using the *Oa* genome instead of the *On* reference genome. Using WGS data from genomic libraries of three tilapia species, which have been reported to segregate for SD on LG1, we investigated the SD critical region. Only one cross-species coding variation determining maleness was observed, in the form of the *figla-like* gene, which is in the orthologous SD region of LG1 on the *Oa* genome. Different cichlid species and tilapia commercial stocks were tested to validate the WGS data results and to investigate whether *figla-like* segregates as an SD MKR in purebred tilapias and hybrids.

## 2. Results

### 2.1. Comparison of the SD Region of LG1 in As, Cz, and Sm

To analyze the critical SD region of LG1, we aligned the WGS data to *Oa* genome (Genome accession number: GCA_013358895.1). We used six genomic libraries of strains and species, which were found previously to segregate for sex on LG1. These included *As*, *Cz*, and *Sm* pools of females and males. Following the Genome Analysis Toolkit (GATK) best-practices workflows of the Broad Institute, we applied short variant discovery within the sequence reads of these genomic libraries and recorded the variants in a Variant Call Format (VCF) text file (Appendix A). For the critical sex region (25.4–28.7 Mbp on the *Oa* map, Appendix A), we obtained 66,372 variants before filtering. Six sites of sequence variation were concordant with the XY model, for all three analyzed fish species (Table 1).

Two sites of variation fitting a model of a null allele on the “X” chromosome were mapped to a predicted *figla-like* gene (*LOC116310109*, positions 26,490,716 and 26,490,863). Another three sites of sequence variation that are in accordance with a heterozygous state in males (XY) and a homozygous state in females (XX) were mapped to the intergenic regions that separate the *figla-like* gene from its two predicted gene neighbors, *CUB and sushi domain-containing protein 1* (*csmd1*) and *chitin synthase 1* (*chs1*) (Figure 1). At position 25,672,475, an additional heterozygous three-allelic variation in males was mapped to an exon of *DEP domain containing 7*, *paralog a gene* (*depdc7a*). However, these genotypes displayed synonymous changes, and although the variation varied between males and females for all species, the male genotypes were not conserved across species.

Visualization of the alignment with Integrative Genomics Viewer (IGV) (Appendix A) revealed that the region of *figla-like* was male specific, and had no aligned reads in females. Thus, we designated *figla-like* as “y” in these species, and its absence as “x” (lowercase distinguishes the xx/xy SD system on LG1 from the *On* XX/XY SD system on LG23). Using the Gap5 software, we assembled the whole gene sequence of the *figla-like* gene for *As*, *Sm*, and *Cz* (nucleotide accession numbers: OX031319, OW742804, and OW742498); this had a three-exon genomic organization, in keeping with the predicted reference transcript (Table 2). The predicted proteins from our assembly contained 99 amino acids and shared a 45-amino-acid homologous basic helix–loop–helix (bHLH) domain with the *Figla* gene from different fish and mammalian species (Figure 2a).

### 2.2. Expression of the Figla-like Gene

Pooled male (*n* = 58, SRA accession number: SRX727305) and pooled female (*n* = 33, SRA accession number: SRX727306) fish from *As* (45 days posthatch) have been previously used for comparing expression in gonads between sexes [32,33]. We aligned the expression data (100 bp reads) from males and females to the *Oa* predicted *figla-like* gene using the Gap5 wrapper and the BWA program [68] and found that this gene was exclusively expressed in males. The average read coverage for this alignment was 86-fold, and the RPKM value was 7.6. We also found *figla-like* gene expression in the expressed sequence tag (EST) database. Expression (>99% identity) was observed in the tilapia adult testis library (2 ESTs, GR703512, GR699597) and in the tilapia adult stomach library (2 ESTs, GR695460, GR693262). These ESTs indicated that the 3′ end of the *figla-like* transcript is longer than was predicted (Table 2).

### 2.3. The Figla-like Gene Is Male-Specific in Different Purebred Species and Hybrids of Tilapia

As the male specific *figla-like* gene was found on LG1 of the *Oa* genome, and it was absent from the *On* genome, we compared the LG1 of both species and developed a duplex PCR-based assay, which detects both forms, i.e., LG1y and LG1x (Appendix A, Table 3). The LG1y marker spanned the *figla-like* second intron, whereas the LG1x marker amplified the orthologous position in *On*, which lacked the *figla-like* gene (Table 3, Appendix A). Using this assay in different *On* strains (*On* Swansea and Ghana) and in *Oa* samples (*Ein-Feshkha* strain), we validated that the *On* PCR product designated as “x” (LG1x) was only amplified in *On*, and that the *Oa* PCR product designated as “y” (LG1y) was only amplified in *Oa* (Table 4). Sanger sequencing was used to validate the amplified fragment sequence origin. In addition, we tested this assay in females and males from three samples of two additional species, *OmI* and *Sg*, which are known to segregate for SD on LG1 (for *Sg*, LG1 SD is reported in paragraph 2.5) [54,62,63]. Validated by Sanger sequencing, a fragment homologous to *On* LG1x was found in all samples, whereas the LG1y fragment was found only in males of both species (Table 4). In two *OmI* families, complete concordance of sex with the xx/xy model was observed. In *Sg*, a single discrepancy of a male lacking a *figla-like* gene was found (Table 4). Using the LG1y/LG1x probe sequences, we further confirmed the male specificity of the *figla-like* gene in pooled male and female samples of *Cz* and *As* (Table 4). In addition, we tested *figla-like* concordance with sex in a family of the *Chitralada* strain (*Cs*), which is a hybrid of at least three species, *Oa*, *On*, and *OmII* [64]. This family’s sex was partially explained by the segregation of the SD locus on LG1 using the microsatellite marker BYL018 and the *figla-like* assay with complete linkage between the two. Only two females had an xy genotype, whereas all other 23 samples segregated for sex according to the xx/xy model (Table 4).

### 2.4. Origin Validation of Species and Strains by Cox1 Sequence

To confirm the origin of species and strains of each library, we assembled the *cox1* barcode sequence, which is the standard for species identification using the BOLD system. The *As* barcode was identical to the barcode of *Oa* (Figure 2c), which suggested a hybrid origin of *As*, explaining the LG1 SD system segregation in this strain despite its annotation as *On* by depositors. The barcode of the *Sm* libraries (SRA accession numbers: SRX1740812 and SRX1740810) had only 0.32% difference from our *Sg* samples. Thus, we concluded that it is likely that these *Sm* libraries have been misidentified and are in fact *Sg*; hence, it is *Sg* and not *Sm* that segregates for the LG1 SD system. In addition, we analyzed barcodes from other *Sg* libraries (SRA accession numbers: SRX9968999, SRX4456733, SRX4456732, SRX4456729, SRX4456726, SRX4456723, SRX4456721, SRX4018194, SRX4018193, and SRX4018191). In these ten libraries, we detected four variants of barcodes with a maximal difference of 0.81% between them, which does not exceed the expected interspecies threshold [44,45,46,47,48,49]. Indeed, one of the barcode variants was identical to the misidentified *Sm* library. Including two resources [69,70] with a similar barcode (differences < 1%) in GenBank, the comparison of the trusted *Sm* barcode sequences showed that the difference between the *Sg* and *Sm* barcodes was more than 4.5%. This further supports the misclassification of *Sg* as *Sm*.

### 2.5. Figla-like Gene and Barcode Sequence Comparison

Using the WGS data and Sanger resequencing of the *figla-like* gene (Table 3), we assembled the whole *figla-like* sequence (Table 2) and compared the predicted protein sequences of the *figla-like* gene for the different species (Figure 2a,b). The *Figla-like* protein sequence was conserved, and there were only a few differences between cichlid species. *Pm* was the most divergent species, having five variations from its closest species *Ot* (Figure 2a). We observed four distinct *Figla-like* protein groups (G), i.e., G1: *Sg*, *Cz*; G2: *Om*, *Oh*; G3: *Oa*, *As*, *Cs*, *Ot*; and G4: *Pm* (Figure 2b,c). The members of each group had identical proteins, except for *Ot* from G3, which differed from the other members of this group by the number of aspartic acid residues in a polyaspartate position (Figure 2a). As the differences on the protein level were low, we compared the nucleotide sequences of the *figla-like* gene. Yet, the differences within the different groups were still negligible. Within groups 1 to 3, the differences in the nucleotide sequences did not exceed 0.18%, 0.07%, and 0.18%, respectively. However, without *Ot*, the nucleotide differences of the G3 members, i.e., *Oa*, *As*, and *Cs*, did not exceed 0.11%. This is in line with the hypothesis that based on the *figla-like* sequences, *As* and *Cs* originated from *Oa* following hybridization. Between groups (G 1 to 3), the nucleotide differences were 0.5–1.5%, indicating high sequence conservation of the *figla-like* gene. The two members of G1 are relatively distant species according to classical taxonomy. Indeed, the distances of G1 based on the *figla-like* gene and barcode sequences were contradictory (Figure 2c). According to the barcode sequence, *Ot* was closely related to *OmI* and *Oh*, whereas their *figla-like* genes clustered in different clades (Figure 2c). In addition, the phylogenetic tree of barcodes showed a complex situation for *Oa*, which did not cluster with other members of the *Oreochromis* genus and seemed to be closer to *Sarotherodon* (Figure 2c).

### 2.6. The Figla-like Gene in Sarotherodon and Coptodon

Among 37 libraries of species from the *Sarotherodon and Coptodon* genera, which were deposited in GenBank (Appendix A), we only found the *figla-like* gene in *Sg* and *Cz*. Using both a 147 bp probe representing Exon 2 from *Sg* and *Cz* and the NCBI BLASTN algorithm, we did not find hits in 22 *Sarotherodon* and 15 *Coptodon* genomic libraries, even though some of these libraries were referred to as males (SRA accession numbers: SRX7645639, SRX7645637, SRX6434288, and SRX6435742) (Appendix A). We only detected the *figla-like* gene in a *Sarotherodon lamprechti* library (SRA accession number: SRX4456739). However, assembly of the barcode and the *figla-like* gene of this library confirmed it was, in fact, an *Sg* sample.

## 3. Discussion

In this study, we observed that the absence of the *figla-like* gene is concordant with femaleness across cichlids with an LG1 SD system, including *OmI*, *Sg*, *Cz*, and certain families isolated from the commercial *As* and *Cs* hybrid stocks. In gonads of these *As* families, we found *figla-like* gene expression exclusively in testes. The *figla* gene has a germ cell-specific basic helix–loop–helix (bHLH) domain, and it is a known vertebrate ovarian factor required for ovarian follicle formation [71,72,73,74]. In mice, *figla* simultaneously suppresses testicular genes and activates many oocyte genes [75,76]. The dimorphic regulation of *figla* is critical for the formation and maturation of primordial follicles. Moreover, similar findings have also been observed in *On*, where *Figla* plays an essential role in the development and maintenance of the ovary and in suppression of spermatogenesis [77,78]. However, in this study, we showed that in the abovementioned cichlids, the bHLH domain containing the *figla-like* gene was involved in male determination. Indeed, it has already been shown, for SD based on *dmrt1*, that the sex-specific paralog may have an opposing function in the determination of male or female sex [10,79,80,81]. A possible opposite function has also been suggested in *Oa* for *banf2w*, which is a paralog of *banf2* [12], and for the *figla* paralogs in tongue sole (*Cynoglossus semilaevis*) [82]. As in tongue sole, the *figla-like* gene may be involved in regulating the synthesis and metabolism of steroid hormones, which are required for male determination. However, the *figla-like* sequence is capable of encoding a relative short peptide, and thus is unlikely to perform complex functions as *figla* does; yet, it is possible that it can drive sex by regulating or competing with its *figla* paralog (Figure 3). Thus, the *figla-like* genomic absence is compatible with a female developmental program, whereas its presence interrupts female development, thus determining male fate. This strongly indicates that the *figla-like* gene is the long-sought SD MKR on LG1.

The discovery of the *figla-like* gene as a candidate MKR for SD was made feasible by using the *Oa* reference genome instead of the *On* genome, which has been used in previous studies, but lacks the *figla-like* gene. As many as 66,372 sequence variants were obtained using the short variant discovery pipeline, which was performed on the LG1 SD’s critical region using meta-analysis of WGS from multiple cichlids. Our strategy assumed that the causative variant was conserved in all cichlid species, which segregate for the LG1 XX/XY SD system. The criterion of conservation across species narrowed the search for the causative sequence to six variants, localized in or in the vicinity of the *figla-like* gene (Table 1).

Examination of the genetic relationship between the different cichlid species was based on the *figla-like* sequence or the mitochondrial barcode. The *figla-like* sequence was consistent with the genus definition of *Oa*, as it groups separately from *Sarotherodon*. However, *Sarotherodon* and *Coptodon*, which are very distant according to their barcode sequences, had similar *figla-like* sequences. According to its barcode sequence, *Ot* is in close relation to *OmI* and *Oh*. However, *Ot*’s *figla-like* sequence clustered it with *Oa* (Figure 2b,c). Surprising results for barcodes have already been shown previously [47,50]. It would be expected that *Oa*, which belongs to the *Oreochromis* genus, would cluster in a phylogenic tree with other *Oreochromis* species; nevertheless, it clustered with *Sarotherodon*. It was also noted that, in some instances, clustering by barcodes seemed to be more consistent with the common geography [47]. Barcode sequences revealed that *OmI* (Appendix A) and *OmII* were different species that were mistakenly referred to as one. Although barcodes only reflect maternal genetic contribution, in this study, they revealed erroneous definitions and faulty origins of species. However, it is puzzling that distant cichlid species segregated for a similar MKR for SD (*Sg* and *Cz*, or *Oa* and *Pm*), whereas closely related species such as *Oa* and *On* segregated for others. It is possible that during speciation there were gene flows between species before mating barriers were fully established [85]. Such flows of the *figla-like* gene or of the mitochondrion might break their genetic linkage explaining the inconsistency between phylogenetic trees generated by their sequences.

In previous studies [12,14], we concluded that three different LGs, i.e., 1, 3, and 23 are involved in the SD of *Oa*, *On*, and their hybrids. Of these, LGs 3 and 23 segregate in purebred *Oa* and *On*, respectively. In the present study, we indicate a candidate gene for a third MKR for SD, *figla*-*like* on LG1 (Table 5). We assumed that the *figla-like* gene is associated with SD following hybridization in two hybrid strains (*Cs* and *As*) and is presumably also an original MKR for SD of three purebred cichlid species (*Cz*, *Sg*, and *OmI*). The hybrid origin of *Cs* and *As* strains was supported by the fact that *Cs* stock included three types of mitochondrial barcodes that were identical to those of *Oa*, *On*, and *OmII* [64], and that the *As* strain carries an *Oa* barcode, even though it has been referred to as *On* by depositors (SRA accession numbers: SRX726489 and SRX726488). Moreover, the sequence of the *figla-like* gene of *As* and *Cs* was highly similar to that of *Oa*, thus indicating the possible role of hybridization in the creation of the MKR. The involvement of genes in tilapia SD has been previously predicted to occur only after hybridization by the autosomal theory [30,38]. This simplistic polygenic theory is able to explain most of the experimental results, assuming that sex is determined by the sum of the effects of three alleles (W, X, Y, where Y = Z) of a major sex-determining locus and two alleles (A, a) of an autosomal locus. The original *Oa* and *On* homozygous states are designated as “aa” and “AA”, respectively, and affect SD following hybridization but not in in the purebred species. We preferred to use “x” and “y” and not “aa” and “AA” following the proposed involvement of the *figla-like* gene as the MKR on LG1 in purebred *Cz*, *Sg*, and *OmI* (Table 5). Thus, here, our findings are explained by a monofactorial system in *Oa*, *On*, and *OmI* and other cichlids. Even though some exceptions have been found suggesting multiple SD systems in a single species [40], they should be treated with caution in view of admixture in aquaculture [86,87,88,89].

As *On* lacks the *figla-like* sequence in LG1, hybrids of the first generation between homogametic purebred *Oa* males (yy/ZZ/XX for LGs 1, 3 and 23) and *On* females (xx/ZZ/XX) result in AMP with a uniform genotype (xy/ZZ/XX), thus being heterozygous only for the *figla-like* gene on LG1 (Table 5). Furthermore, according to this minimal genetic model, *On* xx/ZZ/XX carriers were females as the MKRs for the three SD loci (*figla-like*, *banf2w*, and *amhΔY*) were missing; thus, the developmental program of the female was not altered. In hybrids, this genetic model predicts three possible combinations in each of the three SD loci that form 27 possible genotypes. This may explain why it is difficult to restore AMP production using admixed parental stocks in the absence of an effective assay for genotyping all SD loci. Moreover, the definitions of XX/XY or WZ/ZZ systems are viable only for monofactorial SD systems. Here, we provide a valuable assay that allows simple detection of all three possible genotypes of LGs 1, 3, and 23 for SD in multiple species (Appendix A). The reliability of this molecular assay stems from the cross-species sequence conservation.

Knockout of *amh* in *On* by CRISPR/Cas9 confirmed its involvement in *On*’s SD [36]. In zebrafish, disruption of the *figla* gene by CRISPR/Cas9 led to an all-male phenotype in the mutant [90]. Thus, our suggested candidate gene for the MKR for the SD for LG1, i.e., *figla-like* could be further validated using transgenic fish manipulated by genomic editing with CRISPR/Cas9 or by other methods such as TALEN and antisense RNA [36,91,92,93].

## 4. Materials and Methods

### 4.1. Fish

The purebred *Oa* specimens from local natural resources (*Ein-Feshkha* nature reserve), and the *On* specimens from different introduced strains (Ghana and Swansea) have been previously described [14]. The families of *OmI* were reared in the Volcani Institute of Agriculture from fish, which originated from Natal (RSA) [65]. *Cs* and the families used in this study were reared in the Dor Research Station from a stock described previously [64]. *Sg* specimens were reared in the Ginosar Research Station, which has a breeding stock of *Sg* used for populating the Sea of Galilee on an annual basis [94]. *Cz* samples were retrieved from a previous study [47].

### 4.2. Comparison of the SD Region among On Amherst, Cz, and Sm Strains

The following genomic libraries were used for alignment of the WGS data to the LG1 of *Oa’s* genome map (Genome accession number: GCA_013358895.1) and for variant calling: *On* pools of females and males, designated SRA accession numbers: SRX726489 and SRX726488, respectively; *Cz* pools of females and males, designated SRA accession numbers: SRX3638079 and SRX3638078, respectively; *Sm* pools of females and males with SRA accession numbers: SRX1740812 and SRX1740810, respectively. These alignments and variant callings were performed using best practices of GATK4 [95]. The resulting VCF (Appendix A) was filtered for variants, which could fit three different models of an XX/XY SD system in all three species: (a) variants that were homozygous in females and heterozygous in males, representing a “Y” chromosome that had a different allele to that of the “X” chromosome; (b) variants that were missing in females and were homozygous in males, representing a locus that was absent from the “X” chromosome; and (c) variants that were homozygous in males and heterozygous in females, representing a locus that was absent from the “Y” chromosome. In addition, the SD region for analysis was based on mapping by previous studies between 24 and 27 Mbp in *On* [42,58,60,61], which is orthologous by synteny to the respective region 25.4 and 28.7 Mbp in *Oa* (Appendix A).

### 4.3. Assembly of the Figla-like Gene and Barcode Sequences in Different Species

Assembly of the whole *figla-like* gene was performed for all male libraries described above and for a male library of *OmI* (SRA accession number: ERX3541585), which were all validated by assembly of their barcode sequences. In addition, we assembled the genomic nucleotide sequences of *figla-like* in cichlid fish that segregated a WZ/ZZ system on LG3: *Ot* (SRA accession number: SRX6434465), *Oh* (SRA accession number: ERX4446013), *Oa* from *Ein-Feshkha* (SRA accession number: ERX2240357), and *Pm* (SRA accession number: SRX3638080). We also assembled the barcode sequence of these libraries to verify their origin. The Gap5 [68] wrapper and the BWA program were used for alignment of reads against the predicted *Oa figla-like* gene (*LOC116310109*) and the barcode of *On*, respectively. A consensus sequence was predicted using read pairs data and was used for further BLASTN-searching, obtaining and aligning the reads, until the complete gene structure was constructed. To deduce the *figla-like* exon–intron borders, we used the mRNA data of *As* and assembled the *figla-like* transcript from males (SRA accession number: SRX727305). We also tested the alignment of *As* females (SRA accession number: SRX727306) to negate the expression of the *figla-like* gene in females.

### 4.4. Amplifying and Resequencing of Figla-like and LG1x Sequences

Based on the assembled *Oa* and *Sm figla-like* sequence (which was later confirmed as *Sg*) and the LG1x sequence from the *On* genomic map (Genome accession number: GCA_001858045.3), using Primer3 [96], the primers were designed to resequence the full or partial *figla-like* gene, the LG1x fragment in samples of *Sg*, *Cz* (*Cz* samples from a previous study were not phenotyped for sex), *OmI*, *Oa*, *On*, and the *Cs* hybrid strain, and for testing association between the *figla-like* gene and sex in families of *OmI*, *Cs*, and *Sg* specimens (Table 3). PCR was performed using MyTaq™ HS Red Mix (Bioline Ltd., London, UK) according to the manufacturer’s instructions under the following conditions: 36 cycles for 30 s at 94 °C, 30 sec at 60 °C, and 60 s at 72 °C. Thereafter, the PCR products were separated in a 1–2% agarose gel stained with ethidium bromide. Following excision from the gel, Sanger sequencing was conducted from both directions of the purified products (Montage Gel Extraction, Millipore, Bedford, MA, USA). The diagnostic markers used for testing association (Table 3) were sequenced in the same way for at least two individuals of each species/strain to validate the PCR results. The microsatellite marker BYL018 was used for genotyping the *Cs* family as previously published [19].

### 4.5. Sequence Alignments and Phylogenetic Analysis

The protein sequences were aligned with ClustalW (http://clustalw.genome.jp, accessed on 10 July 2022), using the default settings and the BLOSUM matrix. The graphical image of the multiple alignment was made using BoxShade (https://manpages.ubuntu.com/manpages/jammy/man1/boxshade.1.html, accessed on 10 July 2022). Phylogenetic trees of the *Figla-like* predicted-protein sequences and the *cox1* genes for different specimens were generated by MEGAX [66] using the Maximum Likelihood method and JTT matrix-based model. A discrete Gamma distribution was used to model evolutionary rate differences among sites with 5 categories (+*G*, parameter = 0.2071), and bootstrap analysis (500 replicates) was performed after alignment using MUSCLE. The model was chosen based on the comparison of Bayesian information criterion (BIC) levels for different models, using the MEGAX find best model option. Similarly, the *cox1* phylogenetic tree was generated using the Maximum Likelihood method and the Hasegawa–Kishino–Yano model [67].

### 4.6. Electronic PCR

Using nucleotide probes (Appendix A), we conducted a BLASTN search in GenBank with a 64-word size against SRA libraries (Section 4.2). The number of hits was documented, and for expression analysis, RPKM was calculated (Appendix A). A minimal limit of at least three reads was set as the detection threshold in genomic libraries. Following detection in the genomic libraries, the electronic PCR results for the *figla-like* gene (Table 4) were based on the number of samples reported by the depositors for each of the male and female pools described Section 4.2.

### 4.7. Statistics

The JMP^©^ statistical package (Pro 13, SAS Institute, Cary, NC, USA) was used for conducting Fisher’s exact test, which was applied for testing the association of the *figla*-*like* gene and sex.

## Figures and Tables

**Figure 1 ijms-23-07636-f001:**
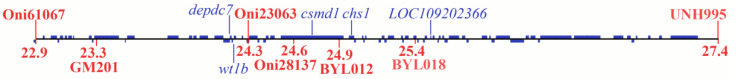
Genetic markers (red) and genes (blue) previously linked with sex on LG1. Positions (Mbp) of genes and genetic makers on the current *On* genomic map build (Genome accession number: GCA_001858045.3) are denoted. Gene symbols of genes mentioned in this article are indicated below or above the bars that delineate their positions.

**Figure 2 ijms-23-07636-f002:**
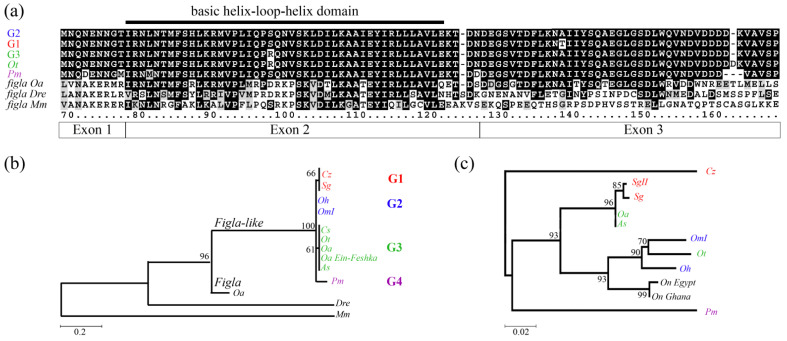
Protein sequence and phylogenetic tree of the *figla-like* gene. (**a**) An alignment of the predicted proteins encoded by *figla-like* and *figla* proteins. The alignment includes five shorter polypeptides, which are of *figla-like* protein groups (G1–4). Protein group G1 (blue) consists of the identical polypeptides of *C. zillii* (*Cz*) (Nucleotide accession number: OW742498) and *S. galilaeus* (*Sg*) (Nucleotide accession number: OW742804). Protein group G2 (red) consists of the identical polypeptides of *O. urolepis hornorum* (*Oh*) (Nucleotide accession number: OW740593) and *O. mossambicus* (*OmI*) (Nucleotide accession number: OW739941). Protein group G3 (green) consists of the identical polypeptides of *Chitralada strain* (*Cs*) (Nucleotide accession number: OW739608), *O. aureus* genomic build (*LOC116310109*), *O. aureus* from *Ein-Feshka* (*Oa Ein-Feshka*) (Nucleotide accession number: OW770257), and *Amherst strain* (*As*) (Nucleotide accession number: OX031319). *O. tanganicae* (*Ot*, green) (Nucleotide accession number: OW739839) is shown out of G3, as its sequence differs by an additional D residue. *P. mariae* (*Pm*, purple) (Nucleotide accession number: OW742294) is the only member in G4. The alignment also includes three partial *figla* polypeptides for *Oa* (Protein accession number: XP_039476449.1), *Danio rerio* (*Dre*) (Protein accession number: NP_944601.2), and *Mus musculus* (*Mm*) (Protein accession number: NP_036143.1). Dashes indicate gaps introduced by the alignment program. Identical amino-acid residues in at least four of eight sequences are indicated by a black background. White boxes indicate nonconservative amino-acid changes between the proteins, whereas gray boxes indicate conservative changes. The black line represents the position of a 45-long basic helix–loop–helix (bHLH) domain found in factor-in-the-germline-alpha (*FIGLA*) proteins. The amino-acid numbering follows that of the full alignment of *figla* with *figla-like* genes (Appendix A). Below, exon–intron boundaries are delineated. (**b**,**c**) Comparison between the phylogenetic trees of *figla-like* predicted proteins and barcode *cox1* DNA sequences. The trees were generated by MEGAX [66] using the Maximum Likelihood method using models with the best Bayesian information criterion (BIC) levels and default setting (5 categories, +G, parameter = 0.2071). Numbers at tree junctions indicate the percentage of trees that correspond to the consensus bootstrap tree (500 replicates) using MUSCLE. (**b**) JTT matrix-based model with a discrete Gamma distribution was used to model evolutionary rate differences among the *figla-like* predicted proteins. The scale on the *X*-axis represents the distance in number of amino-acid substitutions per site. (**c**) Hasegawa–Kishino–Yano model discrete Gamma distribution [67] was used to study evolutionary rate differences among the mitochondrial DNA sequences. The scale on the *X*-axis represents the distance in number of nucleotide substitutions per site. Barcode sequences and accession numbers of *O. niloticus* (*On*) of Egyptian and Ghanaian origin and of others are provided in the Appendix A.

**Figure 3 ijms-23-07636-f003:**
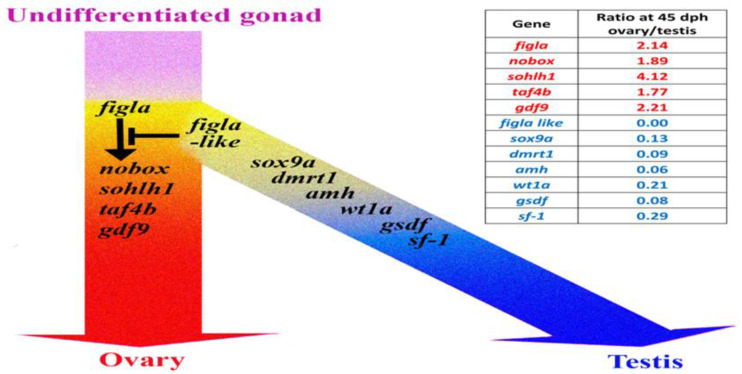
Proposed model of the sex related genes’ expression during LG1-driven gonad differentiation. On the right, the table shows the observed ratio between the RPKM values of these genes calculated for *As* at 45 days posthatch (Appendix A). With increased expression in the ovary (red), *Figla* upregulates a subset of transcripts orthologous to mouse germline genes (*nobox*, *sohlh1*, *taf4b*, and *gdf9*) and their downstream genes, which are essential for early oocyte development [83]. Increased expression in the testis (blue) of a subset of known testis genes (*figla-like*, *sox9a*, *dmrt1*, *amh*, *wt1a*, *gsdf*, and *sf-1*) [84] is compatible with diverting the default female into the male developmental program.

**Table 1 ijms-23-07636-t001:** Sequence variations in the SD region ^1^ on the *O. aureus* genomic map, which fit an XY model in *Amherst strain* (*As*), *S. melanotheron*
^2^ (*Sm*), and *C. zillii* (*Cz*).

Position ^3^	REF ^4^	ALT	*As* F	M	*Sm* F	M	*Cz* F	M	Region	5′ End	3′ End
25,672,475	C	T, G	0/0	0/1	0/0	0/2	0/0	0/1	*depdc7a* exon 8		
26,488,670	T	A	1/1	0/1	1/1	0/1	1/1	0/1	intergenic	*csmd1*	*figla-like*
26,490,716	G	C	./.	0/0	./.	1/1	./.	1/1	*figla-like* exon 2		
26,490,863	C	T	./.	0/0	./.	1/1	./.	1/1	*figla-like* intron 2		
26,509,215	A	C	0/0	0/1	0/0	0/1	0/0	0/1	intergenic	*figla-like*	*chs1*
26,510,329	C	*, T	0/0	0/1	0/0	0/2	0/0	0/1	intergenic	*figla-like*	*chs1*

^1^ The SD interval was chosen for analysis based on synteny with the reference *O. niloticus* genome. ^2^ The *Sm* definition is according to depositors of the library; however, this is challenged in paragraph 2.5. ^3^ in bp on the *Oa* map. ^4^ 0—reference allele (REF, *O. aureus*), 1—alternative (ALT) allele, 2—alternative allele, ./.—null call, M—males, F—females. A, C, G, T and *—adenine, cytosine, guanine, thymine, and deletion, respectively.

**Table 2 ijms-23-07636-t002:** Genomic organization of the *figla-like* gene in *O. aureus*.

Intron ^1^	Exon		Intron	Size
	no.	Size		
…TCCAGCC**ATG**AACC	1	174	TGGAACG**gt**atgta	1290
cttac**ag**ATCAGAA	2	147	TGACAAT**gt**aagta	122
atttt**ag**GATGAAG	3	931	CAGTCCT**TGA**AATG…	

^1^ Intron and exon sequences are written in lowercase and uppercase letters, respectively. The first and last two bases of the introns are presented in bold type (**gt** and **ag** for donor and acceptor splice sites, respectively). The initiation and stop codons are shown in bold and underlined. Considering the predicted transcript (Nucleotide accession number: XM_031726851.2), the genomic size of the *figla-like* gene was 2664 bp.

**Table 3 ijms-23-07636-t003:** Polymerase chain reaction (PCR) primers for generation of amplicons for fragment analysis and resequencing of the *figla-like* gene in *S. galilaeus* (*Sg*) and the *Chitralada* strain (*Cs*).

Marker	Primers		Assay	GenBank Accession	Positions		Amplicon Size (bp)
					Start	End	
LG1y	F	AACCAAGCCAAAATGTGAGC	Duplex fragment analysis	LOC116310109	1520	1821	302
	R	CATTCACTTGCCAGAGGTCA					
LG1x	F	TCTGTGAAGCACTTTGGCATA	Duplex fragment analysis	NC_031965.2	24,979,876	24,980,010	135
	R	CTGCACCTCCTCCAATTGTT					
Reseq1	F	CTTGCACTGGCCTTGAGTTT	Resequencing of *Sg* and *Cs*	NC_031965.2	26,489,072	26,490,461	1390
	R	AAAAATACAGCCAATACATCTGGT					
Reseq2	F	AAAACCAAACAAGGTCACAATTC	Resequencing of *Sg* and *Cs*	NC_031965.2	26,490,237	26,491,052	816
	R	CATTTCAAGGACTGACAGCAA					
Reseq3	F	TGACCTCTGGCAAGTGAATG	Resequencing of *Sg*	ERZ9148259	1556	2526	971
	R	ATGCCTGGACTGGAAACAAG					
Reseq4	F	TGACCTCTGGCAAGTGAATG	Resequencing of *Cs*	NC_031965.2	26,490,991	26,491,772	782
	R	GCCGAGCAGAGCCTAGTTTA					

**Table 4 ijms-23-07636-t004:** Association of sex with the *figla-like* sequence in two *O. mossambicus* (*OmI*) families, *S*. *galilaeus* (*Sg*), *C*. *zillii* (*Cz*), and *Amherst* (*As*) and *Chitralada* (*Cs*) strains.

Species	Genotype ^1^	Females	Males	*p*-Value ^2^
*OmI* Family 1	xy	0	8	0.0002
	xx	7	0	
*OmI* Family 2	xy	0	8	0.0003
	xx	6	0	
*Sg*	xy	0	15	0.0001
	xx	18	1	
*Cz* ^3^	xy	0	9	<0.0001
	xx	13	0	
*As* ^4^	xy	0	58	<0.0001
	xx	33	0	
*Cs* ^5^	xy	2	11	0.0001
	xx	12	0	

^1^ xx and xy genotypes correspond to LG1x/LG1x and LG1x/LG1y, respectively (Table 3, Appendix A). ^2^ Fisher’s exact test. ^3^ Electronic PCR based on pooled samples, SRA accession numbers: SRX3638079 and SRX3638078. ^4^ Electronic PCR based on pooled samples, SRA accession numbers: SRX726489 and SRX726488. ^5^ Identical results were obtained using marker BYL018.

**Table 5 ijms-23-07636-t005:** The schematic allelic state of the SD systems for LGs 1, 3 and 23 in *O. niloticus* and *O. aureus*
^1^.

Linkage Group
	Sex	1	3	23
Species/Proposed SD MKR		*Figla-like* (y)	*Banf2* (W)	*AmhΔY* (Y)
*O. niloticus*	Male	xx	ZZ	XY
Female	xx	ZZ	XX
*O. aureus*	Male	yy	ZZ	XX
Female	yy	WZ	XX

^1^ This table also integrates our previously published results [12,14].

## Data Availability

Data are contained within this article’s Appendix A. Nucleotide sequence data of *figla-like* and *cox1* sequences were deposited in ENA under project accession number PRJEB52443.

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
