# Peer review of "Absence of Figla-like Gene Is Concordant with Femaleness in Cichlids Harboring the LG1 Sex-Determination System"

_ijms, 2022, doi:10.3390/ijms23147636_

Round 1

Reviewer 1 Report

In the present study, the authors identified a male-specific figla-like gene in Coptodon zillii, Sarotherodon galilaeus, O. mossambicus and O. niloticus of the Amherst and Chitralada hybrid stock, harboring the LG1 sex-determination system. The finding is interesting.

Major:

1.        The sex patterned SNPs of As, Cz and Sm spanning from 25.4 to 28.7 Mb on the Oa map should be included as supplementary data, to help understand how the six sites were obtained.

2.        Visualization of the alignment of the male specific region of figla-like of As, Cz and Sm with Integrative-Genomics-Viewer (IGV) should be included in the MS.

3.        Protein sequences of Cz, Oh and Cs were not included in Fig.2a.

4.        Species used for Fig.2a, Fig.2b, Table 4 and Figure S2 should be consistent.

5.        The authors should add some discussion to explain why “distances based on figla-like and barcode sequences were contradictory”.

6.        Line 392, it is difficult to understand why On females (xx/ZZ/XX) was female fish. The authors should discuss more.

7.        Table 5, AmhΔY (Y) should be changed to amhy, as both gain of function and loss of function experiments have proved that amhy is the sex determining gene on LG23 of On.

Minor:

1.        Line 106-108, add references.

2.        Line 220-221, the accession number of expressed sequence 220 tag (EST) database should be shown in the Methods section.

3.        Line 235-236, add references for “OmI and Sg, which are known to segregate for SD on LG1.”.

4.        References format should be checked. For example, Ref.78. “LinYan, Z.; YongXiu, Q.” Should be “Zhou, L.; Qiu, Y.”

5.        Species names in the reference section should be checked. Species name in all Latin names should be in lowercase, while the first letter of the genus name should be in uppercase.

Reviewer 2 Report

Minor editorial revisions will be needed.

1. line 54: Oreochromis niloticus should be O. niloticus

2. line 94: Sarotherodoon melanotheron should be S. melanotheron

3. lines 93, 95, 186, 187, 189, 190, 211: Oreochromis should be O.

4. lines 251-252: O. mossambicus  S. galilaeus  C. zilli  Amhest  Chitralada should be italic

5. line 298: Oreochromis  Sarothrodon should be italic

6. All references: titles should be small capital except the first letter

7. Ref. nos. 4, 13, 26: journal names should be spelled out

8. Ref. no. 11: Mugil Cephalus should be Mugil cephalus

9. lines 553, 568, 570, 573, 595, 597, 604, 608, 611, 615, 618, 623, 659, 662, 668, 671, 674, 677, 689, 724, 757, 761: Oreochromis Niloticus should be Oreochromis niloticus

10. lines 565-566: O. Niloticus and O. Aureus should be O. niloticus and O. aureus

11. lines 575-576: Tilapia Nilotica x T. Aurea should be Tilapia nilotica x T. aurea

12. lines 589: Niloticus  Aureus should be niloticus  aureus

13. line 621: Vol 11, Page 1017 should be 11, 1017

14. lines 629, 639, 725: PLOS should be PLoS

15. lines 656, 699: Melanotheron should be melanotheron

16. lines 689, 764: Mossambicus should be mossambicus

17. line 694: T. aki ?  need to be checked

18. line 699: Melanotheron should be melanotheron

19. line 700: Urophthalmus should be urophthalmus

20. lines 707. 732: Latipes should be latipes

21. line 732: add United States of America after Sciences

22. line 743: Semilaevis should be semilaevis

23. line 760: Esculentus should be esculentus

24. line 781: Lucius should be lucius

25. line 785: Glilaeus should be glilaeus
